# Point-of-Care Ultrasound (POCUS) in Adult Cardiac Arrest: Clinical Review

**DOI:** 10.3390/diagnostics14040434

**Published:** 2024-02-16

**Authors:** Federica Magon, Yaroslava Longhitano, Gabriele Savioli, Andrea Piccioni, Manfredi Tesauro, Fabio Del Duca, Gabriele Napoletano, Gianpietro Volonnino, Aniello Maiese, Raffaele La Russa, Marco Di Paolo, Christian Zanza

**Affiliations:** 1Department of Anesthesia and Critical Care, Bicocca University of Milano, 20126 Milano, Italy; magonfederica@gmail.com; 2Department of Anesthesiology and Perioperative Medicine, University of Pittsburgh, Pittsburgh, PA 15261, USA; lon.yaro@gmail.com; 3Departement of Emergency, IRCCS Fondazione Policlinico San Matteo, 27100 Pavia, Italy; gabrielesavioli@gmail.com; 4Department of Emergency Medicine, Gemelli Hospital, Catholic University of Rome, 00168 Rome, Italy; andrea.piccioni@policlinicogemelli.it; 5Department of Systems Medicine, University of Rome “Tor Vergata”, 00133 Rome, Italy; manfredi.tesauro@uniroma2.it; 6Geriatric Medicine Residency Program, University of Rome “Tor Vergata”, 00133 Rome, Italy; christian.zanza@live.it; 7Department of Anatomical, Histological, Forensic and Orthopedical Sciences, Sapienza University of Rome, Viale Regina Elena 336, 00161 Rome, Italy; fabio.delduca@uniroma1.it (F.D.D.); gabriele.napoletano@uniroma1.it (G.N.); gianpietro.volonnino@uniroma1.it (G.V.); 8Department of Surgical Pathology, Medical, Molecular and Critical Area, Institute of Legal Medicine, University of Pisa, 56126 Pisa, Italy; 9Department of Clinical Medicine, Public Health, Life Sciences, and Environmental Sciences, University of L’Aquila, 67100 L’Aquila, Italy; raffaele.larussa@univaq.it; 10Italian Society of Prehospital Emergency Medicine (SIS 118), 74121 Taranto, Italy

**Keywords:** pulmonary embolism, cardiac arrest, pocus, pneumothorax, emergency medicine, ultrasounds, echocardiography

## Abstract

Point-of-Care Ultrasound (POCUS) is a rapid and valuable diagnostic tool available in emergency and intensive care units. In the context of cardiac arrest, POCUS application can help assess cardiac activity, identify causes of arrest that could be reversible (such as pericardial effusion or pneumothorax), guide interventions like central line placement or pericardiocentesis, and provide real-time feedback on the effectiveness of resuscitation efforts, among other critical applications. Its use, in addition to cardiovascular life support maneuvers, is advocated by all resuscitation guidelines. The purpose of this narrative review is to summarize the key applications of POCUS in cardiac arrest, highlighting, among others, its prognostic, diagnostic, and forensic potential. We conducted an extensive literature review utilizing PubMed by employing key search terms regarding ultrasound and its use in cardiac arrest. Apart from its numerous advantages, its limitations and challenges such as the potential for interruption of chest compressions during image acquisition and operator proficiency should be considered as well and are discussed herein.

## 1. Introduction

Cardiac arrest is defined as sudden loss of heart activity, resulting in ineffective breathing and blood circulation. The swift initiation of Advanced Cardiac Life Support (ACLS) or Basic Life Support (BLS) protocols stands as a crucial element within the chain of survival in cardiac arrest to improve survival.

The delivery of high-quality CPR with minimal pauses between compressions aims to restore blood circulation, while the use of an AED could be used to treat shockable rhythms such as ventricular fibrillation (VF) and ventricular tachycardia (VT). On the other hand, the management of pulseless electrical activity (PEA) and asystole requires prompt diagnostic evaluation of potentially treatable or reversible causes [1].

In this setting, observational research has highlighted the significant role of Point-of-Care Ultrasound (POCUS). Employing POCUS enables precise identification of reversible conditions, thereby offering valuable guidance for ongoing resuscitation efforts.

The POCUS technique’s main limitations have always been identified as its sensitivity and specificity together with its training-dependent variability [2]. On the other hand, its low cost and rapid execution have made POCUS the first-line examination choice not only for in-hospital cardiac arrest. Out-of-hospital cardiac arrest, with an incidence rate of 55 per 100,000 adults and a fatality rate of nearly 90%, is a hurdle that could benefit from greater portable ultrasound availability. Given the broadly increasing application and the possibility to transmit ultrasound images, the introduction of more portable and durable technologies could bring an increase in diagnostic potential and guide therapeutic interventions in the field.

The accessibility and immediate feedback offered by Point-of-Care Ultrasound render it indispensable in emergency and critical care settings. This portable diagnostic tool delivers real-time insights, thereby enabling the rapid and precise assessments that are crucial in high-stakes situations. In recent years, resuscitation guidelines have increasingly recognized the potential role of POCUS in cardiac arrest scenarios and have emphasized its multifaceted benefits, although admitting that its usefulness has not yet been well established [3,4,5,6]. Specifically, ultrasound applications aim to optimize cardiopulmonary resuscitation outcomes by facilitating crucial tasks such as:Diagnosis and subsequent intervention;Identification of reversible causes of cardiac arrest;Monitoring and guidance of advanced resuscitation.

This narrative review summarizes the multifunctional role of sonography during cardiac arrest. We analyze its significance as a component of resuscitation protocols and discuss its diagnostic potential and its additional utility during ACLS. Potential pitfalls and possible strategies to render its application effective in directly influencing patient outcomes are discussed as well.

## 2. Materials and Methods

We conducted an extensive literature review by utilizing PubMed/Medline, Ovid/Wiley, and the Cochrane Library databases, thereby employing key search terms such as “POCUS”, “cardiac arrest”, “cardiac ultrasound”, “emergency medicine”, “cardiac tamponade”, “tension pneumothorax”, “hypotension”, and “pulmonary embolism.” Our search was focused on identifying relevant clinical trials and review articles published within the last two decades, excluding case reports. After meticulous evaluation, we handpicked more than 50 publications that we deemed to be the most pertinent and informative. Additionally, we augmented our findings by including relevant papers sourced from the reference lists of the selected articles. Our aim was to review the latest literature to provide clinicians with current insights into the utilization of bedside echography in emergency settings, specifically during cardiac arrest.

## 3. Discussion

### 3.1. Technique

POCUS can be focused on a cardiac view or it can include other windows. In performing POCUS during cardiac arrest, the ideal probe is dependent on the goal. A phased array transducer, working at low frequencies (1–5 Hz) and providing high resolution, is usually adopted as the cardiac probe. A linear transducer (7–15 MHz) is optimal for the lung window and pneumothorax, thereby allowing via its high frequencies detailed imaging of the pleural line [7,8]. Deep tissue and abdomen evaluation is best carried out with a convex probe characterized by a wide footprint and frequencies ranging from 2.5 to 7.5 MHz. Regardless of the availability of different probe options, it is preferable to mostly rely on one probe, preferably a phased-array probe, during cardiac arrest scenarios. This is because the phased-array probe offers versatility and comprehensive assessment capabilities, thus allowing for efficient evaluation of both cardiac activity and lung function. The main difference between echography during cardiac arrest and standard echography lies in the approach and timing used. Rapid acquirement of a single view is the aim during the critical event of cardiac arrest where precious seconds cannot be spent evaluating multiple measurements and views as is the case with standard echography. With this objective in mind, the subxiphoid approach has traditionally been the preferred window as it can be performed during ongoing CPR. Nonetheless, the parasternal long-axis view demonstrated superiority compared to the subxiphoid view regarding the rapidity of execution as well as the quality of the cardiac image [9] (Figure 1). Ultimately, patient characteristics (e.g., body habitus), as well as the environment and the accessibility to the patient (particularly in a prehospital context), may dictate the best view for each situation [9,10].

POCUS can be utilized during cardiac arrest to provide additional information. For example, placing a probe on the trachea can confirm the correct placement of the endotracheal tube, thus ensuring effective airway management. Through examination of the lower extremities, Doppler assessment can detect potential vascular issues such as deep vein thrombosis, which can contribute to an arrest or complicate the resuscitation process. The aortic artery and the abdomen can also be evaluated in the search for sources of blood loss [3,11]. Ultrasound can also be used for obtaining venous access during cardiac arrest. Performing US-guided femoral catheterization might be faster and more likely to succeed than other approaches, as suggested by a small-scale trial [12]. The application of POCUS during cardiac arrest can also be extended to transcranial Doppler ultrasound. The evaluation of cerebral blood flow could guide external cardiac massage to produce an improved and sufficient blood flow, hence contrasting neurological sequelae [13].

### 3.2. Clinical Validation

Incorporating POCUS into the management of cardiac arrest benefits from the adoption of protocols that provide standardization and guarantee quality and efficacy, thereby limiting evaluations to a maximum pause of 10 s per pulse assessment [14]. Numerous protocols have been proposed for POCUS assessment, largely overlapping and all aspiring to operationalize the sonography approach to cardiac arrest [15,16,17,18,19,20].

The majority of these protocols stem from expert perspectives, with some having undergone assessment through trials. Some of the earliest proposals include the Focused Echocardiographic Evaluation in Resuscitation (FEER) and the Focused Echocardiographic Entry Level (FEEL), both of which focused on TEE and were based on a stepwise approach using four phases repeated in a cyclic manner to continually reassess cardiac status and response to interventions. The phases include: (1) preparation of the team and the equipment, (2) image acquisition taken in advance of a pulse check pause (<10 s), (3) prompt CPR resumption, and (4) image interpretation, communication, and action [20]. The FEEL protocol underwent a prospective, observational trial, showing that focused echocardiography findings altered management in 78% of cases [16]. Encouraged by this study, a series of additional protocols were proposed, widening the scanning area to other anatomic locations. Pulmonary evaluation was introduced in the Cardiac Arrest Ultrasound Exam (CAUSE) whereas an abdominal focus was firstly recommended in the Cardiac Arrest Sonographic Assessment (CASA) protocol [10,15].

Subsequently, the pulseless electrical activity protocol (PEA) (representing Pulmonary, Epigastric, and Abdomen/additional scanning regions, although not to be performed in this order) and the Sequential Echocardiographic Scanning Assessing MEchanism (SESAME) protocol have augmented an expanding array of approaches for employing ultrasound techniques not only in cases of cardiac arrest but also in severe shock [18]. More recent protocols that include multiple sonographic views mostly propose the execution of a cardiac exam to be performed during sequential pulse checks that must be no longer than 10 s apart and the acquisition of additional images during active CPR. In this way, pulmonary, abdominal, and vascular views can be obtained, thus guaranteeing continuity of thoracic compressions. The Sonography in Hypotension and Cardiac Arrest (SHoC) protocol instructs on the evaluation of the four “Fs”: the mandatory Fluid, Form and Function and supplemental views assessing the Filling [19]. Among the most recent protocols, the Echocardiographic Assessment using Subcostal-only view in Advanced Life Support (EASy-ALS) proposes the only cardiac view obtained by residents that undergo structured training [21]. Notably, all existing protocols require a trained physician to perform the examination, and some also include the presence of a timekeeper who counts out loud during pulse checks.

### 3.3. Prognostic Role of POCUS in Cardiac Arrest

Approximately 80% of cardiac arrests that occur within hospital settings consist of rhythms that are not amenable to shock [22]. They include pulseless electrical activity (PEA), pseudo-PEA, and asystole. Both PEA and pseudo-PEA involve the absence of a palpable pulse. PEA specifically denotes the presence of organized electrical activity without effective cardiac contractions. Pseudo-PEA highlights a deceptive scenario where the observed electrical activity on the ECG does not correlate with an actual palpable pulse. Asystole is a state of complete cessation of both electrical activity and contraction. The absence of cardiac activity in PEA portends worse outcomes. Patients in PEA with cardiac standstill on ultrasound have survival to hospital discharge rates ranging from 0.0% to 0.6% [23,24]. On the other hand, pseudo-PEA is linked to a higher probability of achieving return of spontaneous circulation (ROSC) [16]. Thus, differentiation between these rhythms is of prognostic value. Using an electrocardiogram alone can be challenging; Breitkreutz et al. demonstrated that asystole documentation with an ECG alone could be misleading, and up to 35% of patients stated to be in asystole demonstrated coordinated cardiac motion in echo [12]. A prospective, observational trial of patients undergoing prehospital ACLS found that using focused cardiac ultrasound allowed for the identification of pseudo-pulseless electrical activity and other findings that altered management in 78% of subjects [16]. Given this, POCUS detection of spontaneous cardiac movement (SCM) could predict the likelihood of survival and delay the decision regarding the termination of resuscitation [25].

### 3.4. Diagnostic Role of POCUS in Cardiac Arrest

Cardiac arrest necessitates identification of any underlying treatable conditions. The reversible causes of cardiac arrest are organized by the ACLS guidelines into the “Hs and the Ts” (Hypoxia, Hypovolemia, Hyperkalemia/other electrolyte disorders, Hypothermia, Thrombosis, Tamponade, Tension pneumothorax, and Toxic agents) [6]. While some of these can be identified through clinical history and laboratory testing, direct sonographic assessment has been proposed to provide evidence of tamponade, pulmonary embolism, tension pneumothorax, and hypovolemia. In this way, POCUS can provide a diagnosis and guide management, as long as no interruption in resuscitation is determined.

#### 3.4.1. Cardiac Tamponade

Cardiac tamponade is a potentially reversible cause of cardiac arrest. Chest trauma, especially penetrating chest trauma, as well as non-traumatic causes can lead to it. While it is generally understood that large pericardial effusions can lead to tamponade, it is important to note that its physiology can result from effusions as small as 50 mL.

The use of sonography can help diagnose tamponade within minutes. Signs consistent with tamponade include the presence of fluid between the fibrous and serous pericardium, the collapse of the right cardiac chambers during diastole, and a small ventricular size (Figure 2; Figure 3). An exaggerated drop in systolic blood pressure due to impaired filling of the heart might also be identified with a paradoxical shift of the septum during inspiration. A dilated and poorly collapsible inferior vena cava may be observed as well [26]. Prompt intervention for tamponade can quickly address pulseless electrical activity (PEA), leading to higher survival rates compared to other causes of PEA (15% vs. 1.3%) [5,24]. In the presence of suspected or confirmed cardiac tamponade, an emergency pericardiocentesis to drain the accumulated fluid is indicated to restore circulation. The use of sonography also guides and assists with pericardiocentesis, thus enhancing the precision and safety of the procedure, with a success rate >90%. In a study evaluating the use of TTE in patients with cardiac arrest related to trauma, its utilization led to reduced time spent in the resuscitation area. Additionally, a decrease in the necessity for blood transfusions and a reduced need for invasive procedures was shown in these patients, which translated into a lower utilization of resources [27].

#### 3.4.2. Tension Pneumothorax

Lung collapse and the subsequent compression of vital structures leads to decreased cardiac output and impaired blood circulation, ultimately resulting in cardiac arrest. Although clinical evaluation can raise suspicion, it is the sonographic findings that play a valuable role in aiding the rapid recognition of tension pneumothorax and drive appropriate intervention. Chest ultrasound shows high accuracy in pneumothorax diagnosis [28]. Signs suggesting the presence of this condition during cardiac arrest comprise a lack of lung sliding determining an “A profile”, as well as underfilled cardiac chambers and a plethoric inferior vena cava (Figure 4). Conversely, the presence of a lung pulse, determined by the absence of sliding but with synchronous beating of the visceral pleura with the heart, rules out pneumothorax. In severe cases, tension pneumothorax can cause a mediastinal shift, where the heart and major vessels are displaced due to pressure from the collapsed lung. Identifying these sonographic signs of tension pneumothorax during cardiac arrest is crucial for prompt diagnosis and appropriate intervention. Immediate decompression with a thoracostomy to release the trapped air can be performed as an emergency life-saving procedure [29].

#### 3.4.3. Hypovolemia

Cardiac arrest can result from hypovolemia due to decreased intravascular volume and extensive vasodilation. Hemorrhagic shock is the primary cause of death in this setting, especially in trauma patients [30]. While reduction in blood volume might be evident in traumatized patients, detecting the origins of internal bleeding presents a more considerable diagnostic hurdle, which can be addressed by employing ultrasound. The evaluation of size and function of the ventricles can aid in determining whether there is reduced cardiac filling due to decreased blood volume. Reduced left ventricular size might suggest hypovolemia [31]. The left ventricle might be hyperdynamic, with obliteration of the LV volume at the end of systole (Figure 5).

The inferior vena cava (IVC) acts as a vital blood reservoir, holding 85% of the total plasma volume within the venous circulation. Adjustments of circulating volume lead to variations in the caliber of the IVC. Indeed, the finding of a “flat vena cava” (e.g., an IVC with an anteroposterior diameter of less than 9 mm) at multiple levels is associated with significant hypovolemia (Figure 6) [32]. The IVC is usually visualized from a subcostal view via a longitudinal scan, including the veno–atrial junction and the right atrium with the inner walls clearly visible. In case of a suboptimal or unavailable subcostal window, a coronal transhepatic scan along the posterior right axillary line may be an effective alternative. Most authors suggest that measurements should be taken within 1.5 cm from the IVC-to-right atrial junction [33]. However, it is essential to consider that the interpretation of IVC measurements in the context of hypovolemia is not always straightforward. Venous congestion during CPR can limit the significance of the IVC assessment. Other factors such as intrathoracic pressure changes due to mechanical ventilation, individual patient characteristics, and concurrent medical conditions can influence IVC size and collapsibility [34,35].

#### 3.4.4. Pulmonary Embolism

A pulmonary embolism, especially if massive, can explain up to 5% of cardiac arrest cases. Systematic thrombolysis in the context of cardiac arrest resuscitation is controversial, although existing consensus guidelines support the possible use of thrombolytics for confirmed or presumed PE [6,36]. Fast diagnosis of PE as the cause of cardiac arrest can direct thrombolytic therapy, which translates into a significantly higher achievement of ROSC when compared to patients who do not receive it [37]. An echocardiographic examination is suggested by different guidelines for patients with suspected PE, especially if they are hemodynamically unstable [38]. In this population, the absence of echocardiographic signs suggestive of PE exclude it as the cause of hemodynamic instability.

Dilatation of the right ventricle in opposition to a small left ventricle is a typical right heart strain sign that can be seen on an echo and provides clues for the presence of PE. The diagnosis of dilation relies on a comparison between the diameters of the RV and LV at the end of diastole. A normal relationship is defined as 0.6:1 and can be evaluated via an apical four-chamber view, while a ratio of 1:1 is diagnostic of RV dilatation. Nonetheless, the association between RV dilatation and PE can be weak, given the fact that patients with cardiac arrest commonly present RV dilatation, particularly in cases where resuscitation efforts extend in duration [39]. As another sign of right heart strain, the right ventricle can present in a D shape as visualized in a parasternal short-axis view (Figure 7. Less frequently, a clot can be identified as a direct sign of PE [40]. In addition, as mentioned above, some protocols include scanning for deep vein thrombosis as an indicator of pulmonary embolism during arrest [41].

#### 3.4.5. Hypoxia

Hypoxia, frequently accompanied by elevated carbon dioxide levels (hypercarbia) resulting from respiratory conditions, accounts for the majority of noncardiac arrest cases. Cardiac arrest solely due to hypoxemia is infrequent. Pulseless electrical activity (PEA) commonly entails complete airway obstruction and typically manifests within 5–10 min. The underlying factors leading to hypoxia, like pleural effusion or consolidation, can be identified through use of sonography once airway control has been established. Sonography excels at distinguishing pleural effusions by detecting even small amounts: as low as 3–5 mL. In pleural effusion, fluid localizes superior to the hemidiaphragm (Figure 8). As regards to consolidation, it typically presents as a hyperechoic area with air bronchograms, often associated with a loss of normal pleural-line dynamics. A specific sign, a shredded pattern, can be seen. It appears as irregular, fragmented lines resembling shredded paper. This pattern can also become tissue like; the complete loss of aeration permits the transmission of ultrasounds through various structures, rendering bronchi and pulmonary vessels visible (Figure 9).

### 3.5. Other Applications

Identifying the return of spontaneous circulation through pulse palpation may be compromised for various reasons. Factors such as weak and inconsistent pulsations or challenging environmental conditions as well as individual variations can contribute to the unreliability of this method, even when it is performed by trained staff. In the context of cardiac arrest, this could result in compression administration in vain or even erroneous CPR termination. On the other hand, detecting a pulse might indicate ROSC and thus enhance quick reassessment of the patient’s vital signs and overall condition, thereby eventually providing post resuscitation care. The use of cardiac ultrasonography allows for faster and more precise pulse detection compared to manual analysis and Doppler sonography [42,43]. Pulse analysis with POCUS can be performed even faster via individuation of the arteries prior to the pulse check. This method has been demonstrated to be precise and reliable and rapid to execute and learn, especially with proper training and practice [44].

Additionally, POCUS can be used during cardiac arrest to identify inappropriate compressions. Proper compression is pivotal to cardiac arrest management, with it being correlated with ROSC and survival. Correct administration generates a blood flow that is 20–30% of normal cardiac output but should still produce a palpable pulse. Nonetheless, the hands are commonly placed in a position where the area of maximum compression is the aorta in most patients [45,46]. Additionally, as mentioned above, pulse palpation can be difficult and unreliable even for experienced clinicians. In this context, measurement of end-tidal carbon dioxide might offer a more accurate estimation of blood output [47]. Nevertheless, waveform capnography can be unreliable in several situations, and using an approach which is blind can be time-wasting. In this setting, ultrasound can be used during CPR to detect the area of maximal compression (AMC), thus confirming the correct hand position to maximize cardiac output [48]. However, standardization issues are present, and guidance on the variables to be assessed for compression evaluation is lacking.

### 3.6. Limitations

Performing and interpreting POCUS during cardiac arrest necessitates specific skills and expertise. Moreover, the availability of ultrasound machines, probes, and trained personnel may be limited in certain settings, restricting the widespread use of POCUS during cardiac arrest.

The general guidelines for using POCUS during cardiac arrest indicate that its use should not delay or interrupt the delivery of high-quality CPR. Thus, sonography assessment should be performed within the 10 s pulse check pause and should focus on key assessments. It is clear that the main concern when using this tool regards the prolonged interruption to chest compressions, which has a negative impact on survival [49]. CPR pauses should not be longer than 10 s in order to maintain cerebral blood flow. The use of POCUS during cardiac arrest has been related to prolonged pauses in chest compressions by at least two studies [50,51]. It is important to highlight that the cohort studied was limited in number and engaged only a single site, without assessing the technique used and the quality of the image. It was later demonstrated that sonography can be conducted rapidly without prolonging the 10 s pulse check pauses recommended by guidelines [52]. Clearly, to help avoid issues, sonography must be performed by the most experienced clinician, who should not be the team leader [47].

The obtainment of an adequate window, especially in the heart, can be challenging during cardiac arrest, thus representing an important limitation in the use of TTE. Various elements, including the constrained time window during chest compressions, might reduce both the utilization and the image quality of an echo during resuscitation. Therefore, identification of the best cardiac window should happen prior to pausing compressions through the acquisition of brief and poor-quality images [53].

TTE can be ineffective for several reasons such as patient factors and the presence of interfering material. An alternative to TTE has been recognized in focused transesophageal echocardiography (TEE). TEE maintains both the diagnostic and prognostic potentials of TTE but retains some of the advantages. TEE allows for continuous visualization of the heart during cardiopulmonary resuscitation, thus minimizing compression interruptions. TEE can be positioned during active compressions and can allow for continuous monitoring of the AMC. TEE using the four-view approach was shown to have an impact on diagnosis and therapy, thereby guiding clinical management and having prognostic consequences in 97% (32/33) of the patients in a prospective observational study [54]. Moreover, TEE was shown to facilitate the onset of extracorporeal life support (ECLS). Its diagnostic potential is higher than TTE. In a retrospective observational study, 55.6% of the examinations performed in an ED had findings that could not be easily identified with TTE [55]. Focused TEE use is supported by guidelines given its diagnostic ability [56]. However, its use requires specialized training and expertise.

## 4. Conclusions

The use of ultrasound is becoming an essential skill in the management of cardiac arrest. The implementation of ultrasound technology and its accessibility have historically posed challenges in the management of cardiac arrest. Nowadays, most hospitals are well equipped with ultrasound machines, and the use of portable devices is also spreading. This should reduce the time taken for transportation to just a few minutes, during which time initial actions in the management of cardiac arrest like starting compressions or providing early defibrillation can occur while waiting for the first pulse checks, when ultrasound is most useful. Utilizing sonography is part of a comprehensive approach to identify reversible causes and guide appropriate interventions aimed at improving the chances of successful resuscitation. Ultrasound can also help in prognostic evaluation and could help in optimizing chest compressions. The balance between obtaining crucial diagnostic information and ensuring uninterrupted CPR is critical for optimal patient outcomes during cardiac arrest. We discussed different protocols that are available to guide the application of sonography in cardiac arrest assessment. Even though some impact on outcomes has been correlated with their application, further studies on a larger scale are needed. Survival as an outcome measure is influenced by numerous factors, making it challenging to assess. Moreover, demonstrating a correlation with better outcomes when ultrasound is used can be an arduous task when a lack of cardiac activity should indicate the termination of resuscitation. Future randomized trials could focus on determining whether ultrasound has an impact on more specific endpoints, such as identifying the reversible causes of arrest that could lead to therapies that support long-term resuscitation.

Additionally, efforts in training personnel must be taken into consideration to improve the outcomes of POCUS assessment. Despite the challenges it may present, the latest ACLS guidelines officially recommend the use of focused ultrasound in cardiac arrest. With sufficient training and precautionary measures, POCUS can and should be integrated into the management of cardiac arrest.

## Figures and Tables

**Figure 1 diagnostics-14-00434-f001:**
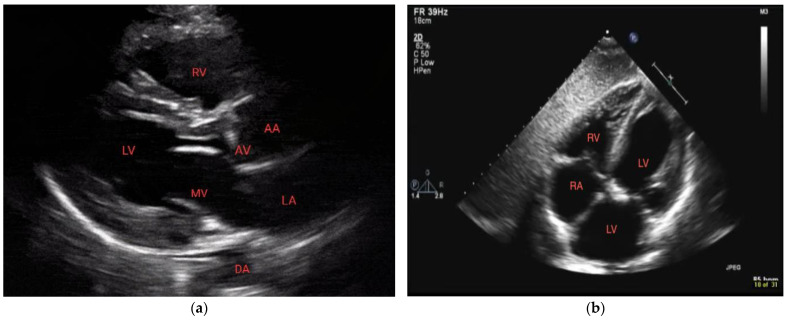
Normal parasternal long-axis view (**a**) and subxiphoid view (**b**). RV = right ventricle, LV = left ventricle, MV = mitral valve, DA = descending aorta, AA = ascending aorta, and AV = aortic valve.

**Figure 2 diagnostics-14-00434-f002:**
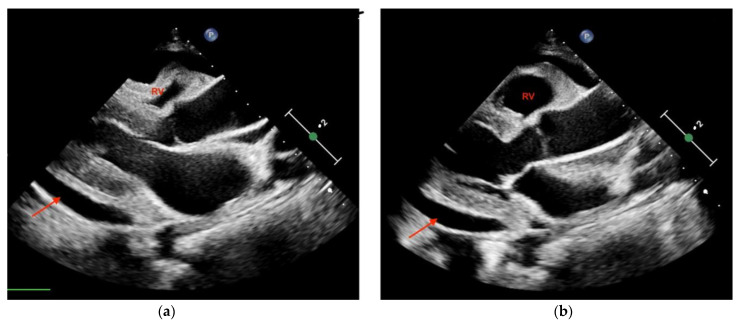
Transthoracic echocardiography (PSLA view) showing the anechoic separation of the pericardial layers measuring 1.4 cm in diastole (arrow). (**a**) Right ventricle (RV) systolic collapse is shown on the left image; (**b**) right ventricular outflow tract diastolic collapse can be seen on the right image.

**Figure 3 diagnostics-14-00434-f003:**
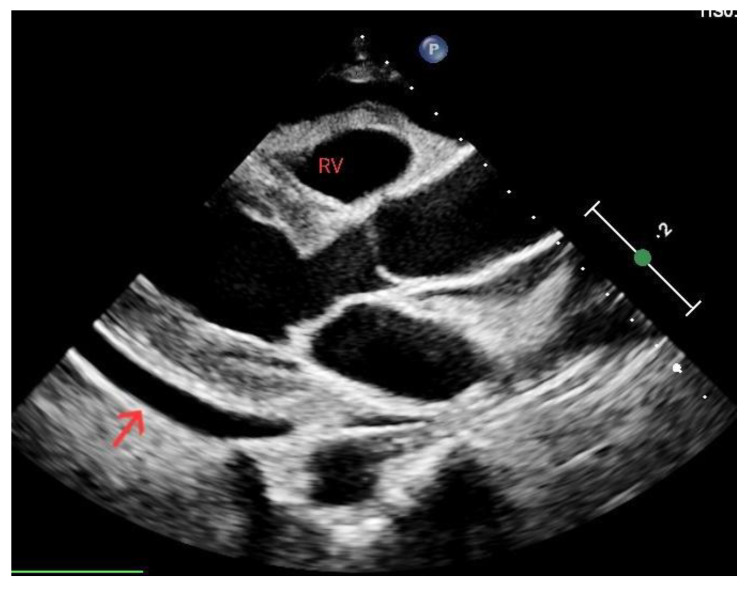
Parasternal long-axis view of a pericardial effusion as indicated by the arrow. The right ventricle (RV) is not collapsed; tamponade is not diagnosed.

**Figure 4 diagnostics-14-00434-f004:**
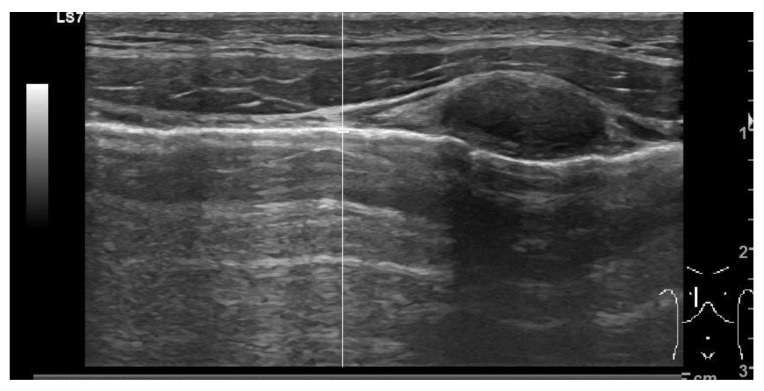
Ultrasound showing pneumothorax resulting in an “A profile”: horizontal artifactual repetitions of the pleural line.

**Figure 5 diagnostics-14-00434-f005:**
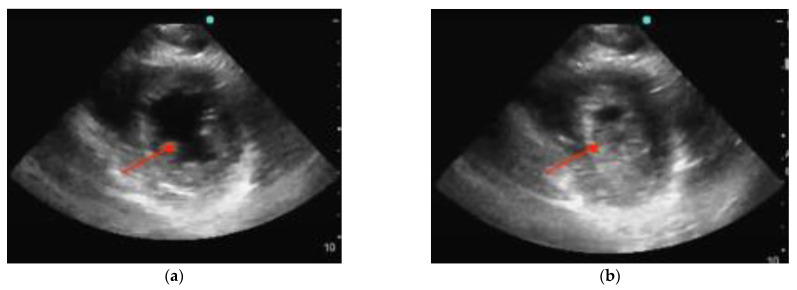
Left ventricle (arrow) during diastole (**a**) and at the end of systole, obliterated and suggesting hypovolemia (**b**).

**Figure 6 diagnostics-14-00434-f006:**
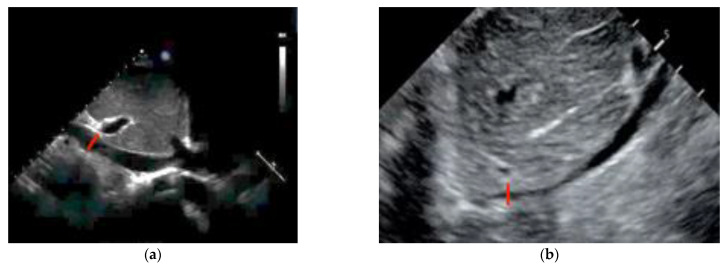
Normal (**a**) vs. collapsed IVC (**b**) indicated by the red lines, suggesting hypovolemia.

**Figure 7 diagnostics-14-00434-f007:**
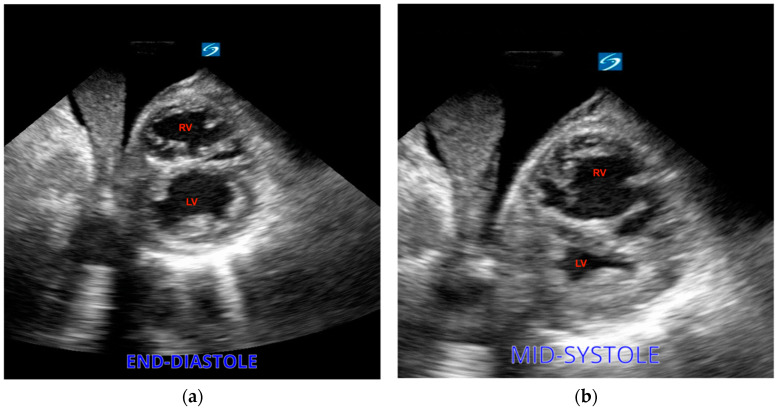
(**a**) Severe right ventricular dilation showing a D-shaped RV. (**b**) The left ventricle (LV) is underfilled.

**Figure 8 diagnostics-14-00434-f008:**
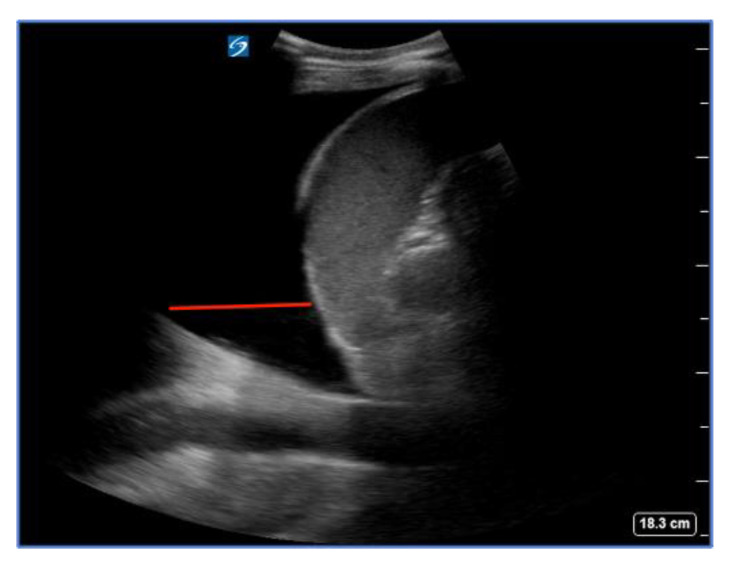
Large volume anechoic effusion visualized superior to the diaphragm and spleen on a bedside ultrasound image.

**Figure 9 diagnostics-14-00434-f009:**
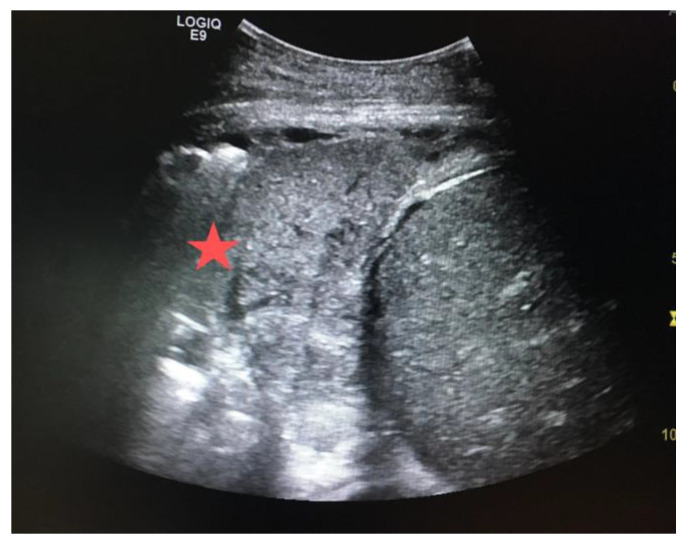
Tissue-like pattern identifying a consolidation.

## Data Availability

Not applicable.

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
