# Peer review of "Point-of-Care Ultrasound (POCUS) in Adult Cardiac Arrest: Clinical Review"

_diagnostics, 2024, doi:10.3390/diagnostics14040434_

Round 1

Reviewer 1 Report

Comments and Suggestions for Authors

Comments on the Quality of English Language

Author Response

First of all, thank you for your thorough review and valuable suggestions. We've carefully incorporated them, resulting in a revised text that is now more streamlined and concise. As suggested, the link to the forensic aspect has been ignored, rendering the review lighter. We reviewed the introduction, leveraging on the three major axes that were mentioned: diagnosis, identification of reversibile causes of cardiac arrest, monitoring and guidance of resuscitation. The text was streamlined omitting technical details on how to perform the exam, deleting part of the TEE paragraph and the discussion on interaction with the family. We created a separate paragraph specifically addressing protocols, as recommended.  All missing references have been added and images have been made clearer by adding annotations such as arrows. The mentioned inaccuracies were revised: clots are in fact rare to find and the mention to hyperkinesia on figure 8 as well as systole-diastole times referred to general evaluation on motion echocardiography: it was excluded due to its lack of clarity. In the limitations paragraph we did briefly discuss technique issues, dedicated training and experience, feasibility in prehospital settings and in the Emergency Department (ED). A few new references on pulmonary embolism were added.

Reviewer 2 Report

Comments and Suggestions for Authors

Magon and Co-authors presented a review on the use of point-of-care ultrasound in cardiac arrest. This is an interesting topic related to the difficulty of collecting data in a such critical setting.

Some minor issues are present.

Some figures' numbers are repeated.

In figure 1, some abbreviations are not listed.

Other Figures are without legends and I suggest to add abbreviations and their explanations

In the lung figure 3, the "lung point" is not showed. It seems a lung point in the m-mode part of the screen but the B-mode seems to be a lung area and a rib. I suggest to change the figure with a more explanatory.

In the Hypovolemia paragraph as well in the pulmonary embolism one, there are an "(a)" and a "(b)" in a row, it seems a typo.

At the beginning, the Authors refers to the lung ultrasound to the use of linear probe, then they show images acquired with the conves probe. It could be great if they will better explain the use of both probes in the lung setting.

I am not sure about figure 7. It is difficult to think to a pneumonia with this image. I suggest to replace it with a better image.

Comments on the Quality of English Language

English language is adequate

Author Response

Thank you for the valuable feedbacks. Figure 1 has been revised, annotations for figures have been included, and we have updated Figure 3 to provide a clearer depiction of the "lung point." We checked and didn’t notice any typo as in the downloaded version “(a)” and “(b)” are below the corresponding images. Nonetheless, we hope any possible typo as been corrected with the revised version. Regarding Figure 7, we understand your concern about the clarity of the image in relation to pneumonia. After careful consideration, we thoroughly reviewed available options, but unfortunately, we couldn't find a more suitable image that effectively captures the specific sign being conveyed.

Reviewer 3 Report

Comments and Suggestions for Authors

In the section 3.3.5. Pulmonary embolism the authors should refer to the amended ESC and ACC guidelines about acute pulmonary embolism, where the echocardiographic algorithm, which is important for emergency diagnosis, is extended and defined. The main changes in recomendation from 2014. are directed in addition to RV dilatation to RV function and level of RV systolic pressure with ultimate goal of improving risk stratification of hemodynamic unstable patients with acute PE. (European heart J 2020, 41: 543- 603)

Comments on the Quality of English Language

quality of English Language is very good. 

Author Response

Following your valuable suggestions, we have reviewed the amended ESC guidelines regarding acute pulmonary embolism and we have incorporated the signs mentions on the guidelines into the Pulmonary Embolism section (3.3.5). We couldn’t find tailored algorithms on cardiac arrest but we did add the recommendation to perform sonography on unstable patients suspected for PE as mentioned in the guidelines.  We are confident that our revised content aligns with the latest recommendations, providing a more comprehensive overview of the topic.

Round 2

Reviewer 1 Report

Comments and Suggestions for Authors

Author Response

We checked every manuscript-line and addressed any corrections according your revision.

Thank you for your suggestion now it is more fluent and easy for readers.